# Wound-Related Complication in Growth-Friendly Spinal Surgeries for Early-Onset Scoliosis—Literature Review

**DOI:** 10.3390/jcm11092669

**Published:** 2022-05-09

**Authors:** Michał Latalski, Grzegorz Starobrat, Marek Fatyga, Ireneusz Sowa, Magdalena Wójciak, Joanna Wessely-Szponder, Sławomir Dresler, Anna Danielewicz

**Affiliations:** 1Childrens’ Orthopedic Department, Medical University of Lublin, 20-093 Lublin, Poland; michall1@o2.pl; 2Childrens’ Orthopedic Department, Children University Hospital, 20-093 Lublin, Poland; starobrat@o2.pl (G.S.); marekfatyga@umlub.pl (M.F.); 3Department of Analytical Chemistry, Medical University of Lublin, 20-093 Lublin, Poland; ireneuszsowa@umlub.pl (I.S.); magdalenawojciak@umlub.pl (M.W.); slawomirdresler@umlub.pl (S.D.); 4Sub-Department of Pathophysiology, Department of Preclinical Veterinary Sciences, Faculty of Veterinary Medicine, University of Life Sciences, 20-033 Lublin, Poland; joanna.wessely@up.lublin.pl

**Keywords:** surgery, QOL, children, surgical site infection, scoliosis, complication, wound, healing

## Abstract

Background: The treatment for early-onset scoliosis (EOS) is one of the most challenging for pediatric orthopedics. Surgical treatment is often necessary, and wound problems and surgical site infections (SSIs) are common, with potentially severe complications in these patients. The aim of the study was to review current literature according to this complication. Methods: PubMed, Cochrane Library, and Embase were systematically searched for relevant articles by two independent reviewers in January 2022. Every step of the review was done according to PRISMA guidelines. Results: A total of 3579 articles were found. Twenty four articles were included in this systematic review after applying our inclusion and exclusion criteria. EOS surgery has a varying but high rate of wound-related problems (on average, 15.5%). Conclusion: The literature concerning the definitions, collection, and interpretation of data regarding EOS wound-related complications is often difficult to interpret. This causes problems in the comparison and analysis. Additionally, this observation indicates that data on the incidence of SSI may be underestimated. Awareness of the high rate of SSI of EOS surgery is crucial, and an optimal strategy for prevention should become a priority.

## 1. Introduction

Early-onset scoliosis (EOS) refers to spine deformity that is present before 10 years of age, with varying etiologies including congenital, neuromuscular, syndromic, and idiopathic. Many EOS patients require surgical treatment, with the insertion of growth-friendly devices, intending to correct the deformity while allowing for continued spinal growth [1]. The most often used are traditional growing rods (TGRs), vertical expandable prosthetic titanium ribs (VEPTRs), magnetic controlled growing rods (MCGRs), and growth guidance systems such as Shilla. Repeated surgeries and complications are two major concerns in the management of EOS. Wound-related problems with surgical site infections (SSIs) are one of the potential complications. It requires a modification of the treatment approach, prolongs the treatment, adds healthcare costs, and causes substantial stress to patients and their families [2]. The presence of SSI is inseparably connected to proper wound healing [3]. The aim of the study was to review current literature to assess the safety of EOS surgical treatment according to the rate of SSI and unplanned surgeries.

## 2. Materials and Methods

### 2.1. Literature Search Strategy

The systematic review was conducted according to the guidelines of the Preferred Reporting Items for Systematic Reviews and Meta-Analyses (PRISMA) [4] (Figure 1). A search of three medical electronic databases (PubMed, Cochrane Library, and Embase) was performed by three independent authors in January 2022. We combined the following terms: “early-onset scoliosis” OR “eos” OR “juvenile scoliosis” OR “infantile scoliosis” OR “tgr” OR “veptr” OR “MCGR” OR “Shilla” OR “growth-friendly” AND “complication” OR “infection” OR “SSI”. The reference lists of all retrieved articles were reviewed for further identification of potentially relevant studies and were assessed using the inclusion and exclusion criteria.

### 2.2. Selection Criteria

Eligible studies for the present review included those dealing with wound-related complications in the operative treatment of Early Onset Scoliosis (EOS) and papers where the main thrust of the paper was not wound problems, though they may have reported it. The initial screening of titles and abstracts was made using the following inclusion criteria: studies in English, reporting clinical results, published in peer review journals, and dealing with SSI complications in operative EOS treatment. When analyzing the papers, it turned out that many of them lack data, so we decided that extracted manuscripts had to contain one of the following: (1) analysis of all four kinds of complications proposed by Bess [5]; (2) analysis of only SSI complication; or (3) papers where wound problems were classified according to the criteria of the Centers for Disease Control and Prevention (CDC) as modified by Horan et al. [6], as superficial or deep infections, and other wound-related problems.

Exclusion criteria were studies with complications in vitro, using an animal model, or dealing with non–operative treatment of EOS. We also excluded all the remaining duplicates, articles dealing with other topics, and those with a poor scientific methodology or without an accessible abstract. Reference lists were also hand-searched for further relevant studies. Reviews, abstracts, case reports, conference presentations, and expert opinions were excluded.

All papers were tagged as follows: (1) the surgical system used: TGR, VEPTR, MCGR, or Shilla (guided growth); (2) number of cases: “big group”—more than 30 cases, “medium group”—10–29 cases, and “small group”—less than 10 cases; and (3) the time of follow-up: “short”—less than 2 years, “minimum”—more than 2 years, and “optimum”—more than 5 years. Final inclusion criteria were primarily limited to “big group” and “optimum follow-up”. During paper extraction, no papers with VEPTR and only one with Shilla and MCGR were found, so those groups’ extracted papers had to be extended with “medium group” and “minimum” follow-up.

### 2.3. Data Extraction and Criteria Appraisal

Three investigators independently reviewed each article. Discrepancies between the reviewers were resolved by discussion and consensus. All data were extracted from article texts, tables, and figures, and put into tables in an excel sheet. As proposed Bess et al., complications were categorized as wound-related, implant-related, alignment-related, and general (surgical or medical) [5]. Surgical procedures were classified as planned (implantations, lengthenings) and unplanned (revisions). Implantation procedures were calculated to equal the number of patients. Not given information was calculated using specific formulas based on the known data, i.e., multiplication of mean number of operations per patient by number of patients = number of operations. Some data—especially in TGR group patients—such as the number of lengthenings and were estimated based on the mean duration between lengthenings using formulas, i.e., multiplication of mean duration between lengthenings by follow-up = the number of lengthenings. Mean duration between lengthenings, if not specified, was taken as a mean value of durations between lengthenings specified in other papers. In some papers, the number of unplanned surgeries wasn’t provided. In those cases, complications as deep infection or implant fracture were estimated as an indication of at least one revision/unplanned surgery. Some fields were left empty when there were not enough data to estimate the value. When the data in the main text and tables didn’t match, the higher value was taken.

## 3. Results

### 3.1. Included Studies

A total of 3579 articles were found. After the exclusion of duplicates, 1452 articles were selected. At the end of the first screening, following the previously described selection criteria, we selected 512 articles eligible for full-text reading. Ultimately, after full-text reading and reference list check, we selected *n* = 24 articles following previously written criteria. A PRISMA flowchart of the method of selection and screening is presented in Figure 1. The included articles focus on complications in the most often used systems: TGR (8 papers), VEPTR (10 papers), MCGR (5 papers), and Shilla (2 papers). One paper described infectious complications in EOS patients regardless of the system (varia). Data extracted from these papers were assigned to the appropriate system.

The papers of Matsumoto et al. [7] and Bachabi et al. [8] compared TGR and VEPTR. Peiro-Garcia et al. [9] compared early complications in VEPTR and magnetically controlled growing rods. As they analyze two systems, these papers appear in the table for the relevant sections.

These 23 studies included a total of 3135 patients; 792 patients were female and 606 were male, and 5 studies did not categorize the patients based on sex. The mean patients’ age at index procedure was 6 years and 10 months. One study did not mention the age at index procedure. The average follow-up was 5 years.

The demographic findings of the included articles are summarized in Table 1.

### 3.2. Patient Demographics

In Table 2, the total number of complications (wound-, implant-, alignment-, and surgical/medical-related) with extracted wound-related complications is presented. As the quantitative data depends on the number of analyzed patients, parts of the table present percentage data. It shows the percentage of wound complication rate per complication, the percentage of wound complication rate per patient, the percentage of unplanned surgeries, and the percentage of unplanned wound surgeries to all unplanned surgeries. In total, 13 out of 24 papers included analyses of all four complications types.

TGRs constitute the most commonly applied technique and are considered the gold standard for EOS with long curves [10]. In the reviewed papers, four out of eight dealt with the TGR system, including implant-, wound-, surgical-, and alignment-related complications [5,8,11,12]. From the group of 412 patients, 291 complications were observed, of which 65 were wound-related. In this analysis, the wound-related complication rate per patient of the growing rod technique ranged from 9.1% to 24.3% (median 12.7%) and wound-related complication rate per complication ranged from 11.9% to 38.7% (median 24.8%). The incidence of unplanned surgery due to wound problems was presented only by Bess et al. [5], amounting to 29 procedures, or 39% unplanned wound-related surgeries in all unplanned surgeries in these patients. These 29 procedures made up 85.3% percent of all wound problems in the group. Two authors focused only on wound-related problems. Kabirian [13] and Dumaine [14], with their groups of total 460 patients, both described 25.9% wound complication rate per patient. A detailed description was presented in five papers [5,7,12,14,15]. In 404 patients, 97 wound problems were observed (24%). SSI occurred in 19.3%. Superficial infections were 39.3%, while deep infections were 60.7%.

VEPTR was developed for patients with thoracic insufficiency syndrome (TIS), but its indication was extended for individuals with EOS at risk of secondary TIS [16]. In the reviewed papers, four out of ten reviewed dealt with the VEPTR system and implant-, wound-, surgical-, and alignment-related complications.

From the group of 109 patients, 136 complications were observed, of which 37 were wound-related. In this analysis, the wound-related complication rate per patient ranged from 8.3% to 69.6% (median 21.7%) and wound-related complication rate per complication ranged from 6.7% to 51.6% (median 23.8%). The incidence of unplanned surgery due to wound problems was presented in two papers [9,17] and ranged from 31.3% to 57.1% (median 44.2%).

A detailed description was presented in seven papers. In 607 patients, 169 wound complications were observed (27.8%). SSI occurred in 85.7%, while 14.3 were concerned with “other wound problems”. SSI was superficial and deep in 37.5% and 62.5%, respectively. Lucas et al. [18] presented 11 SSI in 6 patients (11% of the patients). In 7 cases, infection was bacteriologically documented. The infection rate was 20.3% per patient. In this group, 9 non-infected skin lesions (skin slough and wound dehiscence) occurred in 7 patients (13% of the patients). The rate of skin lesions was 16.6% per patient. This complication was surgically managed in 89% of the cases. Hasler et al. [17] described efficacy and safety of VEPTR instrumentation for progressive spine deformities in young children without rib fusions. In his cohort, 9 out of 23 patients (40%) sustained 16 wound-related complications: 10 skin sloughs and 6 deep infections (5 patients). These 5 patients required 8 wound debridements and 2 temporary implant removals for infections. The risk of complication was 22% (5/23) per IP and 12% (18/149) per expansion procedure.

An interesting analysis was performed by Matsumoto et al. [7]. He compared complications in the treatment of EOS between rib (VEPTR) vs. spine-based (TGR) proximal anchors. In total, 19 out of 76 patients (25%) treated with VEPTR developed wound problems; 15 of them were SSI and 4 were wound dehiscence. Only 1 SSI and 1 dehiscence appeared in the TGR group. They stratified the cohort into those with congenital/idiopathic etiology and those with neuromuscular/syndromic etiology. Surgical site infection was found to be more frequent in rib-based than spine-based groups for neuromuscular/syndromic patients (rib: 13 and spine: 1 vs. 2 and 0, respectively).

Surgical difficulties, as well as the potentially harmful effect of repeated anesthesia, have led to the adoption of magnetically controlled growing rods, and guided growth systems such Shilla. It is known that magnetically controlled growing rods (MCGRs) can reduce the total number of surgeries and have been shown to have a lower risk of SSI compared to TGR [19]. Unfortunately, they still have a similar rate of overall complications, mostly due to implant-related complications. In the reviewed papers, all five dealt with MCGR and implant-, wound-, surgical-, and alignment-related complications [9,20,21,22,23]. From the group of 155 patients, 83 complications were observed, of which 12 were wound-related. In this analysis, the wound-related complication rate per patient of the magnetically controlled growing rod technique ranged from 4.3% to 12.5% (median 6.7%) and wound-related complication rate per complication ranged from 11.8% to 33.3% (median 16.1%). The incidence of unplanned surgery due to wound problems was presented by two authors. Obid et al., presented 1 deep infection which required unplanned surgery [21]. Peiro-Garcia et al. [9] compared early complications in vertical expandable prosthetic titanium rib and magnetically controlled growing rods. In their study, at the 2-year follow-up, the total complication rate was significantly higher in the VEPTR cohort compared with the MCGR cohort (65% versus 13.3%, respectively), but no significant differences were found in infection rates between the VEPTR and MCGR cohorts (10% and 6.7%, respectively). Two patients treated with VEPTR rods suffered wound infections. One patient required 4 surgical debridements, whereas the other was successfully managed with 1 surgical debridement. One patient treated with a single-stage MCGR system required unplanned surgeries (6.7%) due to implant dislodgement at 8 months post-operation. Revision of 1 of the cephalic implants followed by a wound infection required 1 surgical debridement. In their MCGR groups (37 patients), Obid and Peiro-Garcia described 15 complications (40%), with 2 patients with SSI (5.4%) requiring revision surgery. This is the lowest number of SSI in analyzed growth friendly systems. Two authors distinguished the type of SSI in their groups. Lampe et al., observed 2 superficial and 1 deep SSI (12.5% of patients). All these patients required revision surgery [22]. Urbański et al., observed 2 deep SSI in their group [20]. Authors didn’t indicate if revision surgeries were performed.

The Shilla technique guides spinal growth [24]. The technique first corrects the apical deformity towards a neutral alignment; then, the upper and lower growth guidance portions extend into the distal and proximal areas of the curve, using special screws and caps, allowing the rod to slide with growth in a longitudinal direction. Multiple open lengthening surgeries are avoided, as in MAGEC. In the reviewed papers, none included implant-, wound-, surgical-, or alignment-related complications in their analysis. Also, we could not find a paper in which only wound problems were analyzed. The papers of McCarthy et al., and Nazareth et al., described complications taking into account the types of wound-related complications. In the group of McCarthy et al., 6 out of 40 patients (15%) had a secondary infection (wound breakdown), requiring a return to the operating room [25]. One developmentally delayed patient developed wound dehiscence, with a superficial infection after scratching the wound. The remaining 5 patients developed an infection following wound coverage problems over prominent implants. Four patients required implant removal and remained without an implant for a few months to eradicate the infection. Three of them then underwent reinstrumentation (2 with a Shilla construct and 1 with a VEPTR), and the fourth had definitive fusion with instrumentation. In the group of Nazareth et al., there were 4 wound complications (20% of patients), including 3 wound dehiscence and 1 deep infection that required an unplanned revision.

Only six papers exclusively analyzed surgical site infections; these data are presented in Table 3. Two of these six papers [13,14] analyzed complications in patients treated with TGR constructs, and three with VEPTR [26,27,28]. Two of them focused on patients treated with TGR. A multicenter international database was retrospectively reviewed by Kabrian et al. [13]. They identified 70 deep surgical site infection events in 379 patients (18.5%) treated with growing rod surgery, and followed them for a minimum of 2 years. The authors defined deep surgical site infection as any infection requiring surgical intervention (as also described by Garg et al.). Ten patients (2.6%) had a deep surgical site infection before the first growing rod lengthening, 29 (7.7%) had at least one deep surgical site infection during lengthenings, and 3 (0.8%) had an infection after the final fusion surgery.

Three papers were restricted to SSI in VEPTR spinal surgery. Garg et al., conducted a retrospective cohort analysis of variability of surgical site infection patients treated with VEPTR [28]. The authors defined infection as an event that required a return to the operating room for irrigation and debridement. Patients were excluded from analysis if they had had fewer than four total procedures. In the infection cohort, the most common symptoms were wound drainage and dehiscence (41/55, 71%). Other clinical symptoms present were fever (27/55, 49%), pain/tenderness (27/55, 49%), localized swelling/abscess (20/55, 39%), elevated laboratory values (19/55, 32%), redness/warmth (12/55, 22%), and other symptoms (9/55, 15%). The location of most infections was classified as the entire implant tract or unidentifiable (32/55, 58%), with the remaining located at the proximal end of the construct (9/55, 16%), the distal end of the construct (12/55, 22%), or was not reported (2/55, 4%). In this group, the majority of infections were due to Gram-positive bacteria (80%, 44/55), the most prevalent being methicillin-sensitive *Staphylococcus aureus* (45%, 25/55). Seven infection events were reported as having more than one infecting organism. In the authors’ group, nearly 20% of patients who had had at least four VEPTR procedures developed an infection requiring a return to the operating room for treatment. Crews et al., analyzed 326 VEPTR surgeries in 151 patients during the study period. In this group, 26 SSIs (8.0%) were identified in 22 patients. Three patients had multiple infections. Three infections (11.5%) were classified as deep incisional infections, whereas the remaining (88.5%) were classified as organ/space infections. The majority of VEPTR-related infections in this study were caused by Gram-positive infections. *Staphylococcus aureus* was the most common organism, of which the majority were methicillin-susceptible. The last reviewed paper was that of Striano et al. [27], who noted 40 infections in 166 patients (24%) treated with VEPTR.

One paper described the use of Vancomycin powder in the surgical treatment of EOS without division into systems [29]. There were no papers exclusively describing SSI complications in MAGEC and guided growth systems.

Table 4 presents data where the general wound complication was stratified into superficial, deep, and other wound problems. Such a distinction was presented in 16 out of 23 papers, which is why references without differentiation of these types of complications were excluded from the table.

## 4. Discussion

Early-onset scoliosis patients who have failed conservative treatment require operative intervention. Although there has been significant progress in the development and improvement of growth-sparing techniques, the risk of complications accompanying correction surgeries is still high. Many studies agree that in the case of neuromuscular scoliosis, the probability of a complication is 35%, and in the case of EOS, the probability increases to 48% [30]. Surgical site infections associated with pediatric spinal deformity surgery come second after implant-related complications in EOS surgery [31]. Watanabe believed that a patient’s being young at the time of the index surgery significantly reduced the risk of the child developing a significant deformity, the degree of which at the start of the treatment significantly affects the risk of its course [31]. However, there is an inverse relationship between the age of the index surgery and the number of lengthenings in distraction-based methods [32]. Unfortunately, the multiple incisions and the need for repeated intervention lead to high rates of SSI [28]. Deep surgical site infection was defined according to the criteria of the Centers for Disease Control and Prevention (CDC) as modified by Horan et al. [6]. For surveillance classification purposes, SSIs were divided into incisional SSIs and organ/space SSIs. Incisional SSIs were further classified as involving only the skin and subcutaneous tissue (superficial incisional SSIs) or involving deep soft tissues (e.g., fascial and muscle layers) of the incision (deep incisional SSIs). Organ/space SSIs involve any part of the anatomy (organs or spaces) other than the incision opened or manipulated during the operative procedure. Primarily, this review intended to evaluate papers dealing with SSIs. Unfortunately, authors often do not differentiate types of SSI. Additionally, researchers defined SSI in another way for their studies, e.g., as any infection that required additional surgical intervention [28]. Furthermore, authors did not use the specific description of “other wound-related problems”. Some papers, such as those by Du [11] and McCarthy [25], used the term “wound breakdown”, whilst Liang [12] described “painful scar”. This is why further discussion is limited.

As shown in the reviewed papers, wound complications varied according to which system was used and are relatively common; they can substantially impact postoperative morbidity and the cost of care [33]. Bess et al. [5] demonstrated a linear decrease in complication-free rates for each surgical procedure performed; that is, for each surgical procedure performed in addition to the index surgery, there was an increased risk of complication. A patient who had 7 procedures had a 49% chance of having a complication. With 11 procedures, the complication risk increased to 80%. Complication rates increased with the number of surgical procedures. The complication rate was 40% after 2 procedures and increased to 100% for patients who had more than 11 procedures.

Kabrian et al. [13] evaluated the TGR patients, and the prevalence of infection per diagnostic category was 14.7% in the neuromuscular group, 15.9% in the congenital group, 10.2% in the syndromic group, and 2.7% in the idiopathic group. Analysis performed by authors showed that non-ambulatory status increased the prevalence of deep surgical site infection by 2.9 times, and each revision after the initial growing rod surgery and before the first deep surgical site infection increased the risk of infection by 3.3 times. After eight surgical procedures, the risk of deep surgical site infection increased to approximately 50%. Additionally, stainless-steel implants were 5.7 times more likely lead to developing a deep surgical site infection than those with titanium implants. This result is contrary to Dumaine’s research. She and her team noted that SSI can develop more often with titanium implant vs. stainless-steel or chromium cobalt (57% vs. 33 % vs. 10%, respectively) [14]. In the group of Dumaine et al., infection appeared most often in neuromuscular scoliotic patients (43%), followed by syndromic (38%), congenital (14%), and idiopathic (5%). In the group of Poe Kochet et al., 33% of patients developed SSIs [15]. All of them required reoperation, undergoing a total of 33 procedures: 23 surgical site debridements due to deep infection, 2 irrigation and debridement procedures for superficial skin breakdown, 1 flap for skin breakdown, and 7 instrumentation removals. One patient had 17 debridements followed by instrumentation removal. Reinstrumentation was later successful for this patient. Dumaine et al., reported that 26% (21/81) patients developed SSIs: 15 had a single infection, while 6 patients had multiple infections [14]. Of these patients, 90% of patients (19/21) did not require implant removal to clear an infection, but in this group, several patients underwent staged revision, allowing planned new instrumentation to be placed into a cleaner wound bed. In the presented group, five patients underwent temporary instrumentation removal. Four out of five were strategically removed as part of a planned, staged, upcoming final fusion, to ensure a sterile final fusion environment, and one after final fusion. These patients had a transient loss of correction (mean loss of 18.3 degrees) that was regained upon re-instrumentation.

Garg et al., analyzed SSI in patients treated with VEPTR [28]. Although the lowest infection rate was in idiopathic patients at 6%, it was not significantly different from congenital, neuromuscular, or syndromic diagnoses. Moreover, higher systemic disease severity as measured by the American Society of Anesthesiologists Physical Status (ASA-PS) classification also did not correlate with increased infection rates.

Crews et al., tried to identify risk factors for SSIs following VEPTR surgery in children [26]. Comparing demographic and patient-related factors, the authors noted no differences in incontinence status, scoliosis classification, Cobb angle, or apical level. When surgery-related factors were compared, no differences were noted in surgery duration, blood loss, or volume of fluid administered. The only determinant of SSI following VEPTR surgery was the timing of antibiotic administration—the receipt of antibiotic prophylaxis outside of the 1–30 min interval before surgery was an independent risk factor for VEPTR SSI.

Striano et al. [27] noted that lower body weight and non-ambulatory status was significantly associated with infection rate. The authors identified distal surgical sites as being at higher risk for SSI than proximal ones. Furthermore, rib-based distraction device implantation procedures were identified as being at a greater risk for SSI than expansion or revision procedures. Although the authors did not mark what kind of antibiotic prophylaxis was performed, they discussed the differences with the period before standardized use of vancomycin powder. During this period, there was an overall infection rate of 9.9%. After the use of vancomycin powder became standard, there were 518 operations, with an infection rate of 6.2%. Regarding distal exposure areas, after standard use of vancomycin powder, the infection rate dropped from 11.7% to 3.4%. The authors supported the standard application of vancomycin powder to the distal exposure areas. The distal exposure area infection rate was 75% lower after vancomycin began to be used standardly.

The use of vancomycin powder in the surgical treatment of Early Onset Scoliosis was recently analyzed by Dumaine et al. [29] based on a multicenter database for EOS patients. From 104 patients that sustained at least 1 infection after initial guided growth surgery, the authors identified 55 for further evaluation. The etiology of scoliosis was classified as idiopathic for 5 patients (9%), congenital for 10 patients (18%), neuromuscular for 28 patients (51%), and syndromic for 12 patients (22%). There were 2 cases of wound dehiscence (4%), 7 cases of superficial infection (13%), and 46 cases of deep infection (84%). This study of EOS patients undergoing growth-friendly (TGR, VEPTR, MCGR, Shilla) procedures found that use of vancomycin powder was independently associated with increased risk of cultures with no growth. The authors concluded that surgeons and infectious disease physicians should be aware and adjust diagnostic and treatment strategies appropriately.

In the group of Dumaine et al., an infection prevention bundle was implemented for growing spine patients [14]. The protocol included a preoperative *Staphylococcus nares* screen, which was performed at a preoperative assessment visit approximately 2 weeks prior to surgery. If the screen was positive for methicillin-sensitive *Staphylococcus aureus* (MSSA) or methicillin-resistant *Staphylococcus aureus* (MRSA), patients received intranasal eradication 2 days prior, followed by an additional 3 days following the procedure. Additionally, preoperative antibiotics were selected accordingly. Cefazolin was administered for patients with either a negative screen or MSSA and Vancomycin/Clindamycin for MRSA, with standard antibiotic re-dosing [2]. A similar protocol for surgical site infection prevention for pediatric spinal deformity surgery was proposed by Poe-Kochet et al. [34]. Interventions included preoperative nares screening for methicillin-resistant staphylococcus aureus or methicillin-sensitive *Staphylococcus aureus* 2 weeks preoperatively, and treatment with intranasal mupirocin when positive, a bath or shower the night before surgery, a preoperative chlorohexidine scrub, timing of standardized antibiotic administration, standardized intraoperative re-dosing of antibiotics, limited operating room traffic, and standardized postoperative wound care. Such a protocol seems to be very effective and worth considering to implement in all patients operated on due to EOS.

EOS surgery has a varying but high rate of wound-related complications. From the review of 23 papers with 3135 cases, 486 wound complications appeared (15.5%), of which 445 cases of SSI were observed (14.2%), with 10% described as “other wound problems”. Procedures that require open lengthenings (TGR and VEPTR) have the highest frequency of wound complications, especially SSI (19.7%/17.2% and 26.3%/19.7%, respectively). “Closed techniques” such as Shilla and MCGR have a reduced number of SSIs (16.7% and 7.7%, respectively). In these systems, no “other wound problems” were described. The data are simplified and certainly underestimated, due to the reasons described earlier. The rate of wound-related complications might have been higher than reported, as some authors did not report all wound problems.

## 5. Conclusions

Surgical site infections remain a significant concern in pediatric spinal deformity surgery. Current literature regarding spinal infections is abundant but heterogeneous. The number of patients included is limited, and follow-up after infection diagnosis is short. With many authors not differentiating between superficial and deep infections, the literature concerning the collection and interpretation of data regarding SSIs in EOS surgery is often difficult to interpret. This causes problems in the comparison, analysis, and improvement of spine surgery practice. Further work is needed to unify the descriptions and data analysis by different authors. Optimization of the interventions contained within a bundle utilizing larger standardized datasets and investigations regarding individual interventions in specific patient subpopulations are also necessary.

## Figures and Tables

**Figure 1 jcm-11-02669-f001:**
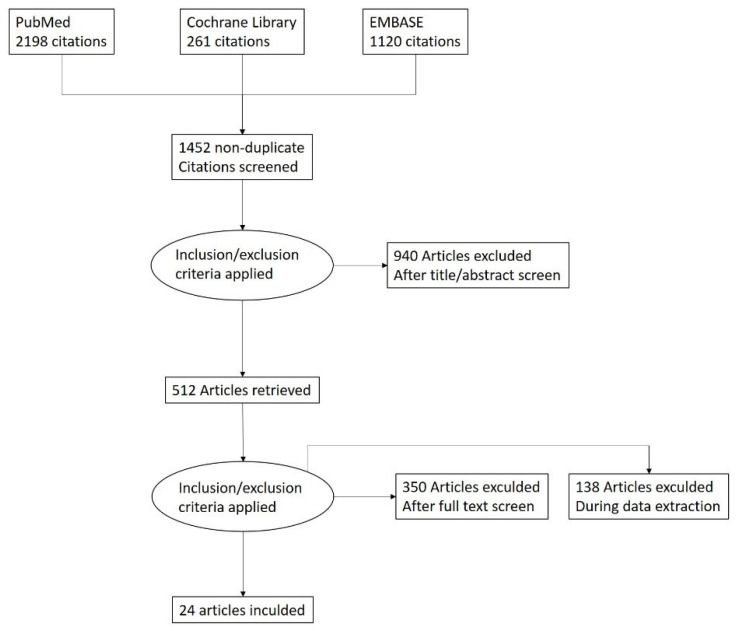
PRISMA flow diagram.

**Table 1 jcm-11-02669-t001:** Demographic data from the reviewed articles. Diagnosis: Neuromuscular (N), Idiopathic (Id), Congenital (C), Syndromic (S), nd—no data. Type of complications: Wound (W), implant-related (I), alignment (A), and medical/surgical complications (M). GGS—Growth guidance system; VEPTR—Vertical expandable prosthetic titanium rib; TGR—Traditional growing rod; MAGEC—MAGnetic Expansion Control.

Ref.	Construct	Subject	Sex (Male/Female)	Age at IP	Follow-Up	Diagnosis	Kind of Complication Analyzed
Bess et al., 2010	TGR	140	71/59	6	5	N (*n* = 52), Id (*n* = 40), C (*n* = 24), other (*n* = 24)	WIAM
Du et al., 2020	TGR	167	69/98	7.2	10.7	Id (*n* = 45), N (*n* = 56), S (*n* = 43), C (*n* = 21), other (*n* = 2)	WIAM
Liang et al., 2015	TGR	55	16/39	6.8	38.4	C (*n* = 28), Id (*n* = 6), S (*n* = 8), N (*n* = 6), miscellaneous disorders (*n* = 7)	WIAM
Poe-Kochert et al., 2016	TGR	100	42/58	7	4.3	N (*n* = 38), S (*n* = 31), Id (*n* = 22), C (*n* = 9)	WI
Kabirian et al., 2014	TGR	379	177/202	6.3	5.3	nd	W
Bachabi et al., 2020	TGR	50	nd	5.5	8.3	nd	WIAM
Matsumoto et.al., 2021	TGR	28	9/19	6.5	5.7	S (*n* = 12), Id (*n* = 5), C (*n* = 1), N (*n* = 10)	WI
Dumaine et al., 2021	TGR	81	30/51	7.3	5	S (*n* = 18), Id (*n* = 19), C (*n* = 13), N (*n* = 31)	W
Crews, 2018	VEPTR	151	16/6	7.1	3+	nd	W
Murphy et al., 2016	VEPTR	25	12/13	5.7	4.5	C (*n* = 25)	WIM
Hasler et al., 2010	VEPTR	23	8/15	6.5	3.6	early onset Id scoliosis (*n* = 1), N (*n* = 11), post-thoracotomy scoliosis (*n* = 2), Sprengel deformity (*n* = 1), hyperkyphosis (*n* = 2), myopathy (*n* = 1), S (*n* = 5)	WIAM
Latalski et al., 2011	VEPTR	12	nd	5.25	2.5	C (*n* = 3), N (*n* = 9),	WIAM
Waldhausen et al., 2016	VEPTR	65	nd	6.9	6.9	C (*n* = 23), N (*n* = 12), S (*n* = 14), Id (*n* = 2), other (*n* = 14)	WI
Striano et al., 2019	VEPTR	166	nd	6.81		N (*n* = 61), S (*n* = 38), C (*n* = 64), Id (*n* = 3)	W
Lucas et al., 2013	VEPTR	54	21/33	7	2	N (*n* = 19), C (*n* = 30), S (*n* = 7), Id (*n* = 3)	WIAM
Peiro-Garcia et al., 2021	VEPTR	20	9/11	4	2+	S (*n* = 5), Id (*n* = 1), C (*n* = 3), N (*n* = 11)	WIA
Matsumoto et al., 2021	VEPTR	76	32/44	6.2	5.7	S (*n* = 11), Id (*n* = 14), C (*n* = 14), N (*n* = 37)	WIM
Garg et al., 2016	VEPTR	38	22/16	5.51	4.1	N (*n* = 18), C (*n* = 13), S (*n* = 5), Id (*n* = 2)	W
Urbański et al., 2020	MAGEC	47	14/18	8.8	1–2.5	N (*n* = 10), S (*n* = 11), Id (*n* = 20), C (*n* = 6)	WIAM
Obid et al., 2020	MAGEC	22	4/18	9.5	3.966667	Id (*n* = 14), neurofbromatosis (*n* = 2), N and S (*n* = 6)	WIAM
Lampe et al., 2019	MAGEC	24	7/17	10.5	3.525	S (*n* = 4), Id (*n* = 9), C (*n* = 1), N (*n* = 10)	WIAM
Lebel et al., 2021	MAGEC	47	12/35	9.2	4.2	S (*n* = 10), Id (*n* = 10), C (*n* = 10), N (*n* = 17)	WIAM
Peiro-Garcia et al., 2021	MAGEC	15	8/7	7	2+	S (*n* = 2), Id (*n* = 1), C (*n* = 3), N (*n* = 9)	WIAM
Nazareth et al., 2020	Shilla	20	10/10	5.7	5.2	S (*n* = 9), N (*n* = 5), Id (*n* = 3), C (*n* = 3).	WIM
McCarthy et al., 2015	Shilla	40	17/23	6.11	5	Id (*n* = 9), C (*n* = 1), N (*n* = 16), S (*n* = 14)	WIA
Dumaine et al., 2021	varia	1115	nd	nd	7.2	no data	W

**Table 2 jcm-11-02669-t002:** Number of wound complications and unplanned surgeries analyzed in extracted papers where additional implant-, alignment-, or medical/surgical-related complications were described.

Ref.	Construct	Subject	Complications Total (No.)	Wound Complications Total (No.)	Unplanned Surgery Due to Wound Problems (No.)	Wound Complication Rate Per Complication (%)	Wound Complication Rate Per Patient (%)	Surgical Procedures (No.)	Planned Surgical Procedures (No.)	Unplanned Surgical Procedure (No.)	Unplanned Surgery in All Groups (5%)	Unplanned Wound Surgery in All Unplanned Surgeries (%)
Bess et al., 2010	TGR	140	177	34	29	19.2	24.3	897	823	74	52.9	39.2
Du et al., 2020	TGR	167	49	19	nd	38.8	11.4	199	167	32	19.2	
Liang et al., 2015	TGR	55	42	5	nd	11.9	9.1	272	263	9	16.4	
Bachabi et al., 2020	TGR	50	23	7	nd	30.4	14.0	179	179		0.0	
Lucas et al., 2013	VEPTR	54	74	18		24.3	33.3	416			0.0	
Peiro-Garcia et al., 2021	VEPTR	20	16	2	5	12.5	10.0		116	16	80.0	31.3
Hasler et al., 2010	VEPTR	23	31	16	8	51.6	69.6	100	86	14	60.9	57.1
Latalski et al., 2011	VEPTR	12	15	1	0	6.7	8.3	183	178	5	41.7	
Urbański et al., 2020	MAGEC	47	17	2		11.8	4.3	50	47	3	6.4	0.0
Obid et al., 2020	MAGEC	22	12	1	1	16.7	4.5	16	12	4	18.2	25.0
Peiro-Garcia et al., 2021	MAGEC	15	3	1	1	33.3	6.7	nd	nd	3	20.0	33.3
Lebel et al., 2021	MAGEC	47	31	5	nd	16.1	10.6	nd	47	nd		
Lampe et al., 2019	MAGEC	24	20	3	3	15.0	12.5	24	24	0	0.0	

**Table 3 jcm-11-02669-t003:** Number of wound complications in extracted papers where only SSI was analyzed.

Ref.	Construct	Subject	Patients with at Least 1 Infection	Wound Complications Total	Infections	Superficial	Deep	Wound Complication Rate Per Patient (%)
Kabirian et al., 2014	TGR	379		70	70		70	18.5
Dumaine et al., 2021	TGR	81	21	27	27			25.9
Crews, 2018	VEPTR	151		26	26	3	23	17.2
Striano et al., 2019	VEPTR	166	40	47				28.3
Garg et al., 2016	VEPTR	213	38	55	55	13	42	25.8
Dumaine et al., 2021	varia	1115	55	55	55	9	46	4.9

**Table 4 jcm-11-02669-t004:** Number of wound complications stratified into superficial, deep, and other wound problems.

Ref.	Construct	Subject	Wound Complications Total	Infections	Superficial	Deep	Other Wound Problems
Bess et al., 2010	TGR	140	30	21	6	15	9
Liang et al., 2015	TGR	55	5	4	2	2	1
Poe-Kochert et al., 2016	TGR	100	33	25	2	23	8
Matsumoto et.al., 2021	TGR	28	2	1	nd		1
Dumaine et al., 2021	TGR	81	27	27	27	0	0
Crews, 2018	VEPTR	151	26	26	23	23	0
Murphy, 2016	VEPTR	25	21	16	8	8	5
Hasler et al., 2010	VEPTR	23	16	16	10	6	0
Waldhausen et al., 2016	VEPTR	65	12	12	3	9	0
Lucas et al., 2013	VEPTR	54	20	9	9		11
Matsumoto et al., 2021	VEPTR	76	19	15	nd		4
Garg et al., 2016	VEPTR	213	55	55	13	42	0
Urbański et al., 2020	MAGEC	47	2	2	0	2	0
Lampe et al., 2019	MAGEC	24	3	3	2	1	0
Nazareth et al., 2020	Shilla	20	4	3	1	2	1
McCarthy et al., 2015	Shilla	40	6	1	1	0	5

## Data Availability

Not applicable.

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
