# Peer review of "Wound-Related Complication in Growth-Friendly Spinal Surgeries for Early-Onset Scoliosis—Literature Review"

_jcm, 2022, doi:10.3390/jcm11092669_

Round 1
Reviewer 1 Report
No doubt there is importance in a comprehensive review and analysis of published literature, as your manuscript does with regard to wound related problems and unplanned surgeries in the treatment of EOS.
There is considerable information in this manuscript, however, my main criticism is the organization and delivery of this data. A literature review should make it easier for the reader to get the salient information that addresses the purpose of the paper. The discussion section is too confusing and essentially is a regurgitation of multiple published data; in other words it needs to be more clearly summarized.
In addition, results are revealed in the discussion section, not where they belong - only tables of data. The discussion should then bring it all together, rather than they reader having to sort through all the papers the authors already reviewed.
I would suggest leaving out any discussion about SSI in conversion of growth friendly techniques to final fusioni since that is not part of this review (e.g lines 316-320).
Table 1: what is “S” - it is not identified in the key? Was it supposed to be “M” ?
please clarify on lines 93 and 96 that * means divided by, or more conventionally / if using a symbol. Also line 204, 222 four “of” eight more conventional than ‘from’.
Finally, additional grammar check required to read better , one e.g. lines 103-107.
Author Response
Dear Reviewer,
we would like to thank you for the helpful comments and suggestions, that improved the quality of our paper. We have revised the paper accordingly and hope that the work is now ready for publication. The changes made are itemized below with our comments (dark blue text) to the reviewer’s suggestions. Changes made in the text are highlighted in yellow in the original manuscript. All the corrections have been made using the “track changes” function of Word.
Reviewer 1
No doubt there is importance in a comprehensive review and analysis of published literature, as your manuscript does with regard to wound related problems and unplanned surgeries in the treatment of EOS.
Thank you for understanding and appreciating the importance of the topic.
There is considerable information in this manuscript, however, my main criticism is the organization and delivery of this data. A literature review should make it easier for the reader to get the salient information that addresses the purpose of the paper. The discussion section is too confusing and essentially is a regurgitation of multiple published data; in other words it needs to be more clearly summarized.
In addition, results are revealed in the discussion section, not where they belong - only tables of data. The discussion should then bring it all together, rather than the reader having to sort through all the papers the authors already reviewed.
Thank you for the comment. Appropriate data were moved to Result section. The discussion was re-edited and clarified.
I would suggest leaving out any discussion about SSI in conversion of growth friendly techniques to final fusion since that is not part of this review (e.g lines 316-320).
Thank you for the comment. The sentence was delayed.
Table 1: what is “S” - it is not identified in the key? Was it supposed to be “M” ?
Thank you for the comment. Letter S (surgical) was corrected to letter M (medical)
please clarify on lines 93 and 96 that * means divided by, or more conventionally / if using a symbol. Also line 204, 222 four “of” eight more conventional than ‘from’.
Thank you for the comment. The symbol * was changed to formula” multiplication of ...by…”. Line 204, 222 – the grammar was corrected
Finally, additional grammar check required to read better , one e.g. lines 103-107.
Thank you for the comment. – Grammar was checked and corrected. The sentence 103-107 was moved to line 70 according to Reviewer 2 suggestion.
Kind regards
Michal Latalski

Reviewer 2 Report
A tabulation of the inclusion criteria questions would be helpful.
Please clarify that you included papers where the main thrust of the paper was not wound problems though may have reported it.
Did you consider index, lengthenings and final fusion separately.
Author Response
Dear Reviewer,
we would like to thank you for the helpful comments and suggestions, that improved the quality of our paper. We have revised the paper accordingly and hope that the work is now ready for publication. The changes made are itemized below with our comments (dark blue text) to the reviewer’s suggestions. Changes made in the text are highlighted in yellow in the original manuscript. All the corrections have been made using the “track changes” function of Word.
A tabulation of the inclusion criteria questions would be helpful.
Thank you for the comment. – Paragraph with inclusion criteria was supplemented, checked and corrected.
Please clarify that you included papers where the main thrust of the paper was not wound problems though may have reported it –
Thank you for the comment. – the proper sentence was added. (line 65).
Did you consider index, lengthenings and final fusion separately.
Thank you for the comment. Separate assessment of index, lengthenings and final fusion would be very interesting, unfortunately was impossible due to the heterogeneity of data in reviewed papers.
Kind regards
Michal Latalski

Round 2
Reviewer 1 Report
Thank you for re-organizing your manuscript. A few questions remain, and several minor editing recommendations:
- In the abstract you refer to a combined wound complication rate and SSI of 15.5% - is this for wound problems or just for SSI’s? I believe wound problem rate was higher. Also, should be ‘on’ average , not ‘one’ (line 25).
- line 71 - delete the ‘s’ after following
- line 166 and line 169 - change ‘from’ to ‘of’
- line 173 - add ‘by’ in front of Bess
- line 179 - SSI in 19.3% (39% superficial and 60% deep). Consider eliminating ‘24% other’, or clarify what this refers to since not clear.
- Line 182 - fix indentation
- line 183, 185 - change ‘from’ to ‘of’
- Paragraph starting at 191 - do the following 2 paragraphs refer to this paragraph? Not clear what goes with lines 191-195ish
- line 207 - ‘An’ interesting…..’was’ by
- line 218 - don’t believe you need new paragraph here.
- line 264 - is there a better place for this paragraph? As one follows the discussion, it seems to be referring to Shilla technique, but I think it is data on TGR. Consider moving to the sections discussing TGR.
- Line 485-486 - varied upon ‘which’ system…..’and’ is relatively common. ‘This complication can’ substantially….
- line 526 - ‘In” the Dumaine et al group, ….
- lines 585-594 - consider making one paragraph.
Author Response
Dear Reviewer,
thank you for the helpful comments and suggestions. The changes made are itemized below with our comments (dark blue text) to the reviewer’s suggestions. Changes made in the text are highlighted in yellow in the original manuscript. All the corrections have been made using the “track changes” function of Word.
- In the abstract you refer to a combined wound complication rate and SSI of 15.5% - is this for wound problems or just for SSI’s? I believe wound problem rate was higher. Also, should be ‘on’ average , not ‘one’ (line 25).
Thank you for the comment. “On” was corrected. Wound complication rate in reviewed papers was 15.5%, It seems to be underestimated as marked in the discussion section.
- line 71 - delete the ‘s’ after following
Thank you for the comment. “S” was deleted.
- line 166 and line 169 - change ‘from’ to ‘of’
Thank you for the comment. “From” to “of” was changed
- line 173 - add ‘by’ in front of Bess
Thank you for the comment. “by” was added
- line 179 - SSI in 19.3% (39% superficial and 60% deep). Consider eliminating ‘24% other’, or clarify what this refers to since not clear.
Thank you for the comment. The unclear part was delated.
- Line 182 - fix indentation
Thank you for the comment. Indentation was fixed
- line 183, 185 - change ‘from’ to ‘of’
Thank you for the comment. “From” to “of” was changed.
- Paragraph starting at 191 - do the following 2 paragraphs refer to this paragraph? Not clear what goes with lines 191-195ish
Thank you for the comment. Paragraphs were connected into one.
- line 207 - ‘An’ interesting…..’was’ by
Thank you for the comment. The sentence was corrected.
- line 218 - don’t believe you need new paragraph here.
Thank you for the comment. Paragraphs were connected into one.
- line 264 - is there a better place for this paragraph? As one follows the discussion, it seems to be referring to Shilla technique, but I think it is data on TGR. Consider moving to the sections discussing TGR.
Thank you for the comment. The paragraph was moved to sections discussing TGR.
- Line 485-486 - varied upon ‘which’ system…..’and’ is relatively common. ‘This complication can’ substantially….
Thank you for the comment. The sentence was corrected.
- line 526 - ‘In” the Dumaine et al group, ….
Thank you for the comment. The sentence was corrected.
- lines 585-594 - consider making one paragraph.
Thank you for the comment. Paragraphs were connected into one.
